# OpenReview forum: "Dynamic Trust Region Adaptation for \\ Human-in-the-Loop Reinforcement Learning \\ in Code Refinement"
_ICLR.cc/2026/Conference — Submitted to ICLR 2026_

### Official Review · Reviewer_EYRa · 2025-10-15

**Soundness:** 3
**Presentation:** 2
**Contribution:** 2
**Rating:** 4
**Confidence:** 2

**Summary:**

The paper introduces a novel dynamic trust region adaptation framework with Bayesian-driven Feedback Confidence Estimator and Adaptive Trust Region Controller. The method combines dynamic trust region strategies with adaptive mechanisms, aiming to address shortcomings in current TRPO algorithms for human-in-the-loop Reinforcement Learning in code refinement tasks. The central idea is to address the variability in human feedback quality by modeling feedback confidence using Bayesian methods and the framework dynamically adjusts policy update bounds and reward shaping based on estimated feedback reliability. The approach is validated through empirical studies in the code refinement environment, demonstrating improvements in convergence stability and final performance.

**Strengths:**

1. The paper is well-motivated. The new trust region strategy and its integration into existing adaptive optimizers show clear contributions.

2. The empirical evaluation is comprehensive and the method performs well on all three metrics when compared with state-of-the-art optimizers.

**Weaknesses:**

1. The accuracy of confidence estimation depends heavily on the initial feature extraction from feedback text. The system's robustness might be compromised when facing adversarial feedback.

2. While the framework claims easy generalization, empirical validation outside code refinement is not provided.

3. The setup for handling conflicting feedback is interesting. However, it lacks a bit of justification.

**Questions:**

1. In the contribution part (line 58), filtering noisy input was mentioned but I could not find this anywhere else. Is this claim well-supported?

2. Referring to W2, are there any adjustments necessary for cross-domain transfer?

---

### Official Review · Reviewer_AENV · 2025-10-27

**Soundness:** 2
**Presentation:** 2
**Contribution:** 2
**Rating:** 2
**Confidence:** 2

**Summary:**

The paper proposes a Dynamic Trust Region Adaptation framework for Human-in-the-Loop Reinforcement Learning in the context of code refinement. The key idea is to dynamically adjust the trust region used in policy optimization based on the estimated confidence of human feedback, which is modeled using a Bayesian Beta-Bernoulli estimator. The system is evaluated on a dataset of 10,000 Python/Java programs with synthetic bugs and style issues, showing consistent gains in code correctness, style improvement, and feedback utilization.

**Strengths:**

Handling noisy, variable-quality human feedback is a major open challenge in HITL-RL, especially in code refinement where feedback can be vague, asynchronous, or inconsistent. The use of a Bayesian confidence model to dynamically modulate trust region size is principled and well-motivated.

**Weaknesses:**

1. Dynamic trust regions based on uncertainty have been explored before (e.g., UA-TRPO), and Bayesian confidence estimation for human feedback is common in preference-based RL. The paper’s combination is sensible but incremental.
2. Human feedback is simulated by injecting noise into labels or scores. No real human annotators are involved in the main experiments.
3. No comparison with recent LLM-based code refinement systems (e.g., Codex), which are strong contenders in this space.

**Questions:**

1. How does DTRA compare against recent LLM-based refinement systems?
2. Can the method scale to real-world pull-request reviews with thousands of lines and multi-file changes?

---

### Official Review · Reviewer_Nu87 · 2025-10-29

**Soundness:** 1
**Presentation:** 1
**Contribution:** 2
**Rating:** 2
**Confidence:** 2

**Summary:**

This paper introduces a dynamic trust region framework for Human-in-the-Loop RL to improve code refinement, addressing inconsistent human feedback quality. It uses a Bayesian estimator to score feedback confidence and an adaptive controller to adjust the policy's trust region, using high-confidence feedback to explore and low-confidence to prevent overfitting.

**Strengths:**

- Adaptive trust region and Bayesian confidence modeling is a creative solution
- The method achieves significant improvements in Code Correctness Score, Style Improvement Ratio, and Human Feedback Utilization.
- The authors explicitly discuss ethical implications (bias, transparency, and privacy in feedback tracking)

**Weaknesses:**

- Generalization Across Domains: While the paper motivates general applicability (e.g., robotics, creative design), all evaluations are in code refinement.
- Limited Exploration of Alternative Alignment Paradigms and Limited Scope of Experimental Baselines
- Dependence on Textual Feature Extraction Quality. If the natural-language feedback is ambiguous, inconsistent, or noisy, the confidence estimates (α, β updates) may be unreliable, leading to suboptimal trust region scaling.
- High Computational Overhead: Maintaining and updating Bayesian parameters for every feedback instance adds significant computational and memory cost.
- Parameter choice and design are not well-explained or are missing supporting evidence to convince that the proposed approach is sound.

**Questions:**

- Why does the paper focus exclusively on PPO-based RLHF when newer, RL-free methods like Direct Preference Optimization have shown improved stability and simplicity? Why not compare with DPO or its recent variants for code refinement?
- The proposed method passively reweights feedback based on confidence. Have the authors considered active feedback selection approaches that query humans only when model uncertainty is high?
- How sensitive is the model to the initialization of the Beta distribution parameters ($\alpha_0, \beta_0$)?
- The clipping parameter $\delta_t$ adapts via tanh($\gamma \theta_t$). How was γ selected, and how sensitive is the training stability to this hyperparameter? (PPO and trust-region methods are highly sensitive to clipping behavior, so understanding how $\gamma$ affects convergence is crucial. The paper provides a default value of 3.0 but does not justify it or analyze sensitivity empirically.)
- How do confidence-weighted rewards (Eq. 13) interact with environmental rewards during early training when $\theta_t$ is unreliable?
- Can the dynamic trust region concept be extended to multi-human settings where different users have independent confidence profiles?

---

### Official Review · Reviewer_3QSA · 2025-10-29

**Soundness:** 2
**Presentation:** 1
**Contribution:** 2
**Rating:** 0
**Confidence:** 4

**Summary:**

Reinforcement learning (RL) from human feedback is a common approach to enhancing code refinement tasks. However, it suffers from the inconsistent quality of human feedback. To address this, the authors propose a dynamic trust region–based RL method that incorporates the estimated confidence of human feedback. The approach consists of three main components:

1. A Bayesian estimator to approximate the importance of human feedback,
2. A trust region controller to adjust policy updates dynamically, and
3. A gated module that selectively incorporates human feedback based on its confidence.

**Strengths:**

\+ Optimizing models with a dynamic trust region based on human feedback confidence sounds like an intriguing idea to explore.

\+ This paper attempts to address a well-known issue in a structured manner.

**Weaknesses:**

Cons:

\- There are too many vague descriptions, which make it difficult to understand the detailed methods proposed, as well as the evaluation results. For instance, it is not explained what "feedback $f_{t+k}$  confirms the original suggestion" means (line 189).

\- Related work is limited and seems not to capture the state-of-the-art developments in RL for coding or RLfH.

\- Numerous grammar errors, format issues, and typos, e.g.," følben" (line 49), having a paragraph with only one sentence (line 104),  "rLO" without definition or explanation (line 117), which largely impacts the readability of this paper. The authors used too many bullet points and enumerated lists to stretch this paper to 9 pages.

\- The paper is addressing a meaningful aspect of model adaptation for coding refinement, but the overall quality is not to the standards of ICLR. The submission reads more like a student project than a polished research paper.

**Questions:**

N/A

---

### Meta-Review · Area_Chair_YJPB · 2026-01-06

**Summary:**

Limited novelty: Multiple reviewers (AENV, EYRa) note that Bayesian confidence estimation and dynamic trust regions are established techniques; their combination, while sensible, lacks sufficient innovation.

Flawed experimental validation: Human feedback is simulated by injecting noise rather than using real annotators, undermining claims about handling real-world inconsistency (Reviewer AENV). Single-domain evaluation: Despite claiming general applicability (robotics, creative design), experiments are limited to code refinement (Reviewer Nu87). Missing comparisons to recent preference-based RL methods and LLM-based approaches.

Critical presentation and writing deficiencies: Multiple grammar errors, typos (e.g., "følben", "rLO"), and undefined terms severely impact readability (Reviewer 3QSA). Core concepts like "feedback confirms the original suggestion" (line 189) lack clear explanation, making methods difficult to parse. Excessive bullet points, single-sentence paragraphs, and poor flow.

**Reviewer Concerns:**

No rebuttals are provided.

**Reviewer Scores:**

All reviewers agree the paper should be rejected.

---

### Decision · Program_Chairs · 2026-01-26

Reject